# Exosomes: Potential Next-Generation Nanocarriers for the Therapy of Inflammatory Diseases

**DOI:** 10.3390/pharmaceutics15092276

**Published:** 2023-09-04

**Authors:** Tosca Mori, Lisa Giovannelli, Anna Rita Bilia, Francesca Margheri

**Affiliations:** 1Department of Chemistry “Ugo Schiff” (DICUS), University of Florence, Via Ugo Schiff 6, Sesto Fiorentino, 50019 Florence, Italy; tosca.mori@stud.unifi.it; 2Department of Neurosciences (Department of Neurosciences, Psychology, Drug Research and Child Health), University of Florence, 50139 Florence, Italy; 3Department of Experimental and Clinical Biomedical Sciences, University of Florence, 50121 Florence, Italy; francesca.margheri@unifi.it

**Keywords:** extracellular vesicles, exosomes, chemical composition, miRNA, nanocarriers, inflammation, neurological diseases, liver, kidney and lung injuries, rheumatoid arthritis and osteoarthritis, intestinal bowel diseases

## Abstract

Inflammatory diseases are common pathological processes caused by various acute and chronic factors, and some of them are autoimmune diseases. Exosomes are fundamental extracellular vesicles secreted by almost all cells, which contain a series of constituents, i.e., cytoskeletal and cytosolic proteins (actin, tubulin, and histones), nucleic acids (mRNA, miRNA, and DNA), lipids (diacylglycerophosphates, cholesterol, sphingomyelin, and ceramide), and other bioactive components (cytokines, signal transduction proteins, enzymes, antigen presentation and membrane transport/fusion molecules, and adhesion molecules). This review will be a synopsis of the knowledge on the contribution of exosomes from different cell sources as possible therapeutic agents against inflammation, focusing on several inflammatory diseases, neurological diseases, rheumatoid arthritis and osteoarthritis, intestinal bowel disease, asthma, and liver and kidney injuries. Current knowledge indicates that the role of exosomes in the therapy of inflammation and in inflammatory diseases could be distinctive. The main limitations to their clinical translation are still production, isolation, and storage. Additionally, there is an urgent need to personalize the treatments in terms of the selection of exosomes; their dosages and routes of administration; and a deeper knowledge about their biodistribution, type and incidence of adverse events, and long-term effects of exosomes. In conclusion, exosomes can be a very promising next-generation therapeutic option, superior to synthetic nanocarriers and cell therapy, and can represent a new strategy of effective, safe, versatile, and selective delivery systems in the future.

## 1. Introduction

In recent decades, nanotechnology has substantially posed the basis to the development of smart nanocarriers, which are currently on the market or under clinical evaluation [1]. These nano-therapeutics represent a unique opportunity to improve the safety and efficacy of conventional drug delivery systems because they can improve drug solubility, enhance its bioavailability, decrease toxicity, and give a superior dose response with consequently greater patient compliance [2].

Particularly, nanomedicine represents a new perspective in the treatment of inflammation, particularly through the development of well-designed nanomedicines, mainly represented by nanoparticles, nanovesicles, and nanomicelles [3]. Therapeutic approaches with nanomedicines are related principally to cancer but also to viral, fungal, bacterial, and parasitic infections. Additionally, nanomedicine is successfully used in other diseases, i.e., cardiovascular, immune, ocular, skin, endocrine, metabolic, blood, and some approaches to pathologies of the central and peripheral nervous systems [4].

Nanomedicines confer distinctive biological properties due to their nano-scale, specific structure, and particular surface properties. These features give many advantages, including increased water drug solubility and stability, improved drug selectivity, controlled drug release—with consequently enhanced bioavailability and therapeutic effects—and decreasing side effects. Their size between a few decades and some hundreds of nanometers is fundamental for a specific phenomenon defined as enhanced permeability and retention (EPR) effect. This effect is a remarkable feature of inflamed tissues, which leads to an increased vascular permeability. Nanomedicines, due to their nanosized structures larger than most drugs, are sufficiently small for extravasation and permeation in the inflamed tissues, but due to the architecture of inflamed tissues, they are submitted to a retention effect and a large concentration accumulation in these specific tissues with, consequently, a very selective targeting specificity. Conversely, to obtain such high concentrations within the inflamed tissue via the EPR effect, nanomedicines should be able to evade the reticuloendothelial system (RES). Indeed, upon intravenous administration, they are rapidly recognized as foreign particles and are opsonized by the adsorption of plasma proteins rapidly eliminated from the systemic circulation. In order to protect nanomedicines from RES opsonization and prolong the half-life of in the circulation, they should be converted to stealth nanoparticles by grafting polyethylene glycol (PEG) residues or polysaccharides onto the nanoparticle surface, imparting a “steric stabilization”, in their interaction with RES. Such nanomedicines are denominated as “long circulating” or “stealth”. In addition to passive targeting, the presence of specific ligands on the surface can selectively enhance the uptake of the developed nanomedicines to specific receptors of cells by active targeting, obtaining a higher specificity and efficacy in achieving the anti-inflammatory effects [3]. Among nanomedicines, vesicles are the most versatile nano-platforms, as reflected by the clinical translation and marketed authorization of numerous formulations [5]. However, the major limitations of these synthetic nano-delivery systems are low physical stability and the moderate ability to proficiently cross the physiological barriers or penetrate the tissues as well as cellular membranes [6].

In recent years, research has focused on other vesicles, the natural cell-derived extracellular vesicles (EVs), as smart nanocarriers for different possible applications because of their unique ability to strongly cross the physiological and pathological barriers and the resulting modifications induced in the targeted tissues and cells [7]. Indeed, EVs are able to cross the blood–brain barrier and other physiological barriers, even if the mechanism is still not completely identified, and they can interact with the cell membrane through a variety of ligands/receptors due to lipid and protein composition. Crossing and targeting properties result in a superior internalization via endosomes with respect to synthetic nanocarriers [8]. According to these data, there are currently several ongoing clinical trials with the aim to evaluate the safety and efficacy of these innovative EV treatments and potentially recompense for the shortcomings of artificial delivery systems, mainly displaying inferior toxicity in the spleen and liver and having a lower immunogenicity [9]. Finally, in order to utilize EVs as drug carriers, a prerequisite is to find a strategy for efficient cargo loading, and two different approaches can be distinguished: an exogenous approach after EV isolation and an endogenous loading can be obtained during the biogenesis of EVs [10].

## 2. Classification and Characteristics of the Different EVs

EVs have a similar architecture to that of synthetic vesicles, such as liposomes, characterized by a basic structure made of phospholipids. Proteins represent the main non-lipid molecules, localized in the intraluminal space or associated with the membrane as unique protein surface decorations, conferring natural targeting properties. Protein composition is related to the different biogenesis pathways, even if it is not very simple to discriminate between the different EV populations [11]. Their inner content also includes sugars, growth factors, protease inhibitors, adhesion integrins, and different kinds of genetic material, including double-stranded DNA and numerous coding and non-coding RNAs [12].

EVs are produced from all types of cells, including prokaryotic and mammalian cells, as evidenced by the analysis of tissue culture supernatants. EXs can be isolated from biological fluids including plasma, saliva, breast milk, cerebrospinal fluids, and malignant ascites. EVs can be distinguished in several subtypes according to their different biogenesis and substantially different size distributions. Generally, three different populations are reported in the literature, namely apoptotic bodies, microvesicles (MVs), and exosomes (EXs), a classification based mainly on their biogenesis, release pathways, function, and size (Figure 1) [7].

Apoptotic bodies have a size from 500 nm up to 5000 nm in diameter, generally tending to be on the larger side, and they are released into the extracellular space via dying cells (Figure 1). Their production is due to the improved hydrostatic pressure during the final phase of apoptotic death. Apoptotic bodies are variable in structure and composition and may contain an extensive variety of cellular components: cytosol portions, chromatin remnants, micronuclei, DNA fragments, degraded proteins, or even intact organelles [13].

Macrophages, parenchymal cells, or neoplastic cells degrade apoptotic bodies after their release in the extracellular space via phagocytosis. Even if there are many reports concerning an apoptotic body formation, studies on their role and function are still very limited [14].

The size of MVs typically range from 100 nm up to 1 μm in diameter and include cytoplasmic material. Their formation is not well understood, and it seems to be related to the donor cell’s physiological state and the microenvironment (Figure 1). Protein and lipid constituents from the cytosolic and plasma membrane are the main constituents of MVs. In particular, protein can have a 100-fold higher concentration in MVs compared to the cell lysate, and these proteins also include cytoskeletal proteins and proteins containing post-translational alterations, such as glycosylation, phosphorylation, and integrins—typically adhesion molecules. Integrins could affect the vesicle trafficking and uptake, while other proteins such as the glycan-binding proteins—generally present on the surface of MVs—could be related to targeting effects [15]. Nucleic acids, and several bioactive lipids, represent other constituents of MVs. Recent studies have demonstrated that the proteomic profiles of MVs are profoundly related to the isolation technique and cell origin. MVs are released within the extracellular space like apoptotic bodies, and once entered into the circulation, they can transfer their cargo to adjacent or distant cells to induce changes in the phenotype or function, which are important for numerous physiological and pathological conditions [16].

Finally, EXs have a diameter ranging from 30 to 150 nm and peculiar characteristics, which can differentiate these EVs from the MVs and apoptotic bodies (Figure 1). EXs are released into the extracellular space through different cells including adipocytes, fibroblasts, erythrocytes, lymphocytes, platelets, dendritic cells, brain and stem cells, and cancer cells. Their biogenesis of EXs occurs by the inner budding of the plasma membrane, generating an endosomal vesicle and multivesicular bodies, which are fused with lysosomes, and generating the EXs into extracellular space (Figure 1). EXs can interact with the extracellular matrix or provoke a reaction within adjacent or distant cells [17].

Most body fluids contain EXs, including plasma and serum, cerebral spinal fluid, synovial fluid, bronchial fluid, saliva, urine, breast milk, semen, amniotic fluid, tears, gastric mediums, lymph, and bile. Their content is strongly related to physiological or pathological conditions, including cancer, neurodegenerative diseases, and viral infections—for this reason, EXs are impressively studied as a source of novel biomarkers (Figure 2). Similarly to MVs, the protein and lipid compositions of EXs are related to their origin. The exosome membrane is made of glycerophospholipids containing long and saturated and unsaturated fatty acyl chains (i.e., phosphatidylserine) and numerous lipid rafts (sphingolipids, cholesterol, and ceramides; Figure 2). The membrane is decorated with different proteins, including proteins that are essential for membrane transport and fusion (annexins, flotillin, and Rab proteins), harbor tetraspanins (CD63, CD81, CD82, and CD9), adhesion integrins, and major histocompatibility complexes (MHC I and II), which are involved in antigen presentation (Figure 2). Inside, EXs contain heat-shock proteins (Hsp60, Hsp70, and Hsp90); proteins associated with MVB biogenesis, i.e., ALG2-interacting protein X (Alix, also known as PDCD6IP) and a tumor susceptibility gene 101 protein (TSG101); cytoskeletal proteins (tubulin, actin, cofilin, profilin, and so on); and metabolic enzymes (pyruvate kinase and GAPDH) (Figure 2). In addition, lipid-layering molecules like glycosylphosphatidylinositol-anchored proteins (LBPA), signal transduction molecules (G protein and protein kinases), and nucleic acids—represented by mRNAs, miRNAs, non-coding RNAs (ncRNAs), and mitochondrial DNA, which are involved in receiving and sending signals in intercellular communications—are also present in EXs (Figure 2) [18,19,20].

EXs have a crucial function in cell–cell interaction by transferring bioactive molecules from the donor cells to the recipient cells thanks to the surface receptor molecules and ligands. For these reasons, EXs possess a unique therapeutic potential for numerous diseases due to this ability and specificity for intracellular transporting [9]. Hence, EXs have been investigated as vectors exerting biological activities in the targeted sites or as nanocarriers loaded with specific molecules to produce a biological reaction [17]. In both circumstances, their isolation and characterization from biological fluids or cell cultures are the critical precondition to establish their therapeutic potential. The internalization of EXs arises via a membrane incorporation route mediated with a lipid raft or via caveolae or clathrin-dependent endocytosis. Phagocytosis and micropinocytosis are also reported as possible techniques of the internalization of EXs [21]. These processes represent therapeutic potential as targeted delivery systems for efficiently performing therapeutic functions in the management of numerous infectious diseases, musculoskeletal and cardiovascular disorders, neurodegenerative disorders, and cancer [9]. Their possible use in different therapeutic applications is also enforced by their very low immunogenicity and their capacity to cross physiological barriers including the blood–brain barrier [22]. About 30 clinical trials have been completed till now and have proved the positive diagnostic and therapeutic effects of exosome-based treatment in various cancers, orthopedic conditions, neurodegenerative and hematological illnesses, and autoimmune and cardiovascular diseases [23].

Numerous studies evidenced how EXs produced by mesenchimal stem cells (MSCs) represent a great potential in mediating the therapeutic functions of these cells, becoming increasingly considered as a novel cell-free alternative to cell therapy with MSCs.

Indeed, MSCs are ubiquitous in numerous tissues, principally bone marrow, adipose, muscle, and bone tissues. MSCs have also been isolated from the brain, thymus, pancreas, liver, spleen, lung, and kidney. Their critical role in creating the tumor microenvironment is also well known, which is fundamental for tumor growth and dissemination. MSCs have an enormous regenerative potency, and for this reason, it is increasing their clinical application in a wide spectrum of diseases. MSCs secrete EVs including MVs, apoptotic bodies, and EXs. In particular, EXs represent the most interesting EVs to be used in clinics because of the advantages of higher safety, convenient storage, transportation, and administration, compared to the transplant treatment of MSCs [24].

## 3. Aim of the Review

The aim of the present review is to highlight the possible role of EXs as nanotherapeutic agents in inflammation, focusing on several different inflammatory diseases, neurological diseases, rheumatoid arthritis and osteoarthritis, intestinal bowel disease, asthma, and liver and kidney injuries. The review would like to emphasize the contribution of EXs as therapeutic agents, both as autologous treatments and carriers of drugs.

Inflammation represents a protective response to pathogenic or injury stimuli to eliminate them, clean dead cells, and start tissue repair; this process is regulated, but it can also be destructive if the inflammatory response is unsolved. EVs and especially EXs have a fundamental role in the initiation, mediation, and resolution of inflammatory diseases. The production of EXs from macrophages induces the production of TNF-α and IL-6, while EXs secreted by mast cells induce the maturation of dendritic cells and assist in the inflammatory response. Activated neutrophils as well as T cells also produce EXs. In addition, immune cells are stimulated from EXs secreted from other cells and activate a pro-inflammatory reaction [25].

Inflammation represents an essential factor in the pathogenesis of numerous diseases, and there is a direct correlation between inflammation and the levels of specific EXs, suggesting that EXs play a significant role in inflammation and immune responses. Growing evidence has definitely suggested the role of EXs in the onset, mediation, and treatment of different inflammatory diseases. Various studies have demonstrated that EXs have potent immunomodulatory activities, mostly through the high expression of miRNAs, which can target the immune system and modulate angiogenesis [26,27,28].

## 4. Studies of EXs in Inflammatory Diseases

### 4.1. Neurological Disorders

Neurological disorders are diseases of the central and peripheral nervous systems and include the brain, spinal cord, cranial nerves, peripheral nerves, nerve roots, autonomic nervous system, neuromuscular junction, and muscles. These disorders comprise Parkinson’s disease (PD), epilepsy, Alzheimer’s disease (AD) and other dementias, and cerebrovascular diseases represented by stroke, migraine and other headache disorders, multiple sclerosis, neuroinfections, brain tumors, and traumatic disorders of the nervous system due to head trauma and other causes. According to WHO, neurological disorders affect hundreds of millions of people worldwide.

It is well known that cells of the central nervous system, namely oligodendrocytes, astrocytes, neurons, and microglia produce EXs, which have an important role as messengers among neural cells both under normal and pathological conditions. These EXs enhance neuronal survival and improve neurite growth by stimulating synaptic activity, increasing ceramide and sphingosine metabolism in the recipient neurons, and releasing synapsin under cell stress or during high neuronal activity [29,30]. EXs are able to mediate neuroprotection, and indeed, astrocytes increase the secretion of high levels of heat shock protein 70 (Hsp70) under oxidative stress and hyperthermia, helping the survival of adjacent neurons. Microglia also produce and release EXs similar to those of B lymphocytes, having a role in the immune system [31,32]. Schwann cells and myelinate axons in the peripheral nervous system produce and release EXs to help axonal regeneration after injuries, which are able to suppress the action of a GTPase (RhoA) activated in response to injury [33].

EXs produced by stem cells can decrease the expression of TNF-α and IL-6 inflammatory cytokines and enhance that of IL-4 and IL-10, reducing brain injury. Furthermore, they can also increase the stimulation of CD4^+^ and CD8^+^ lymphocytes and reduce the number of dendritic cells [34]. In a neuroblastoma tauopathy model, EXs can regulate misfolded protein accumulation and can reduce Aβ deposition in AD experimental mouse models and re-establish the function of microglia, relieving memory deficits. EXs can also transfer phosphorylated tau proteins associated with neurodegeneration, as detected in the cerebrospinal fluid of AD patients [29]. EXs derived from human mesenchymal stem cells (MSCs) show an improvement of the disease in a mouse model of autistic-like behavior in mice, even if the exact mechanism of action is not known [35], while, in a swine model of traumatic brain injury, exosomes originating from human MSCs sustain neuroprotection [36]. An overview of the literature has significantly demonstrated that EXs, particularly those derived from MSCs, neurons, stem cells, microglia, and astrocytes can have a possible role in many neurological diseases, i.e., AD, PD, amyotrophic lateral sclerosis, multiple sclerosis, acute ischemic stroke, and spinal cord injury [37]. Indeed, MSCs can differentiate into different types of cells and produce EXs with potent anti-inflammatory and immunomodulatory activities [38], as well as neuronal stem cells (NSCs) that can differentiate as neurons, astrocytes, and oligodendrocytes [39] and release exosomes, having strong neuroproliferative and neuroregenerative potentials and improving recovery in models of neurological injury [40].

Recently, it has been demonstrated that EXs derived from the oligodendrocyte precursor cells evolving into mature oligodendrocytes showed a substantial efficacy on re-myelination and improvement of motor coordination in model mice [41].

Other cells that are able to generate exosomes with interest in anti-inflammatory and neurogenic properties are pluripotent stem cells [42]. Furthermore, EXs secreted by astrocytes and microglia have an important role in the regeneration and repair of CNS tissues. Indeed, it is reported that EXs from astrocytes stimulate axonal growth and re-myelination and can also restore the spinal cord, while EXs from microglial can control inflammation and stimulate neuronal survival [43,44]. Finally, EXs can also be produced by oligodendrocytes and Schwann cells, first producing myelin and support axons [45], while Schwann cells can restore connections of muscle nerves and tissues [46]. It has been also reported that EXs derived from MSCs have a role in the regulation of both antigen-specific and antigen-nonspecific immunity, thus modulating inflammatory and autoimmune diseases [47].

#### 4.1.1. Alzheimer’s Disease

AD represents the most common type of dementia (accounts for 60–80% of dementia cases) involving the regions of the brain that control memory, behavior, and language. WHO has estimated 47.5 million people, globally, have dementia, with 7.7 million new cases every year. Two kinds of anomalous structures, plaques (deposits of a protein fragment called beta-amyloid) and tangles (twisted fibers of tau proteins) are responsible for damaging and killing nerve cells.

A recent review has reported on the possible role of exosomes from MSCs in AD addressed by four principal actions: release of anti-inflammatory constituents (proteins/lipids/microRNAs/enzymes), immunomodulation, decrease in Aβ plaques via internalization and degradation of Aβ oligomers, and regenerative effects [47]. In vitro EXs derived from MSCs protect neurons from neuronal oxidative stress, suggesting they can have a role in the preservation of synapse integrity when neurons are exposed to soluble oligomers of the Aβ peptide, having a leading role in the cognitive decline in AD. Authors of the study related this activity to the EXs’ content of anti-inflammatory and trophic constituents and antioxidant enzymes [48].

An interesting in vivo study with APP transgenic mice (an AD model) demonstrated that intracerebral injection of neuroblastoma-derived EXs reduced Aβ1-42 by binding to the glycosphingolipids of the EXs’ surfaces and carrying Aβ peptides to microglia, determining their phagocytosis and, as a consequence, their clearance [49]. Using animal models of AD, EXs from MSCs can decrease glial reactivity due to increased levels of the anti-inflammatory cytokines (IL-4 and IL-10) and reduction in the pro-inflammatory ones (IL-1β and TNF-α). Indeed, it is reported that cognitive decline, typical of AD, is related to high levels of TNF-α and IL-1β [50]. From the above studies, it seems that the regulation of inflammatory processes can be related to the behavioral enhancements observed in animal models of AD, suggesting the contribution of iNOS.

In further in vivo studies, using AD mouse models, EXs derived from stem cell decrease A*β* deposition, re-establish microglial function, and relieve memory impairments [51,52,53]. It is reported that the effects of EXs are also related to the regulation of gene expression mainly represented by miRNA (principally miRNA-21), mRNAs, transcription factors, and other non-coding RNAs [47,54].

In conclusions, different groups of researchers have found complementary mechanisms in the possibility of using EXs from MSCs, which could improve long-term potentiation in AD mice [50], increase the expression of synaptic proteins [55], and decrease Aβ olygomers and basal ROS levels in neurons [48].

#### 4.1.2. Parkinson’s Disease

PD is a brain disorder, affecting the nervous system and the parts of the body controlled by the nerves, that causes unintentional or uncontainable movements, causing deep difficulties of coordination and leading to progressive problems of walking and talking. A study reported the exploitability of EXs released from Raw 264.7 macrophages as successful carriers with antioxidant activity. EXs were loaded with catalase, obtaining a high loading efficiency, sustained release, and catalase preservation against protease degradation. Exosomes were readily taken up by neuronal cells in vitro. EXs after intranasal administration proficiently reduced oxidative stress and improved neuronal survival in an in vivo animal model of PD [56]. A further study reported that EXs produced by intracerebrally implanted cells in a mice model of PD decreased neurotoxicity and neuroinflammation through their mRNA cargo [57]. Finally, EXs derived from MSCs and loaded with curcumin significantly improved movement and motor coordination in a PD mice model after intranasal administration. Reported mechanisms include the decreasing α-synuclein aggregates, the promotion of neuron function recovery, and alleviation of neuroinflammation. After treatment with these drug deliveries, the movement and coordination abilities of PD model mice were significantly enhanced [58].

#### 4.1.3. Ischemic Stroke

An ischemic stroke precludes brain cells from getting oxygen and nutrients, and they start to die in a few minutes, resulting in brain damage and other complications. WHO reported that more than 6 million people die because of stroke each year. A recent review addressed the possible regulatory effects of EXs obtained from stem cells on the inflammatory response after an ischemic stroke. EXs derived from MSCs can regulate microglial polarization through various pathways such as NLRP3, NF-κB, and STATs, demonstrating that microglial M1 and M2 phenotypes have a relationship with the inflammatory response after ischemic stroke. In particular, the paper describes micro-RNAs from different origins possibly involved in the effects. It is reported that miRNA21a-5p induces microglial M2 polarization by targeting STAT3, miRNA-138-5p promotes astrocyte proliferation and inhibits inflammatory response, miRNA-542-3p suppresses inflammation and prevents cerebral infarction, miRNA-146a-5p is anti-inflammatory (IRAK1/TRAF6), and miRNA-150-5p reduces neuronal apoptosis and inflammation (TLR5) [34]. A further study demonstrated that EXs originating from adipose-derived stem cells can attenuate stroke. The study showed that exosomes from hypoxic pre-treated ADSCs attenuated acute ischemic-stroke-induced brain injury via delivery of circ-Rps5 and modification of microglia from the M1 to M2 phenotype in the hippocampus [59].

#### 4.1.4. Multiple Sclerosis and Autoimmune Encephalomyelitis

Multiple sclerosis (MS) is a chronic, lifelong autoimmune inflammatory disease that can affect the brain and spinal cord, causing serious disability and a wide range of potential symptoms, including problems with vision, arm or leg movement, sensation, or balance. In MS, the immune system attacks the layer that surrounds and protects the nerves called the myelin sheath, meaning that messages travelling along the nerves become decreased or disrupted. Exactly what causes the immune system to act in this way is unclear, but most experts think a combination of genetic and environmental factors is involved.

A very recent study demonstrated that EXs originating from RAW macrophages and loaded with resveratrol significantly decreased inflammatory responses in both the central and peripheral nervous systems in a mouse model of multiple sclerosis, successfully improving the clinical progression [60]. EXs, when administered intranasally, significantly improved disease severity and decreased inflammatory responses in the central nervous system (TGF-β, IFN-γ, IL-1β, IL-6, and IL-17), in brain/spinal cord (IL-10 in spinal cord), in periphery (TGF-β, IL-1β, IL-6, IL-17, and IL-10), and in spleen/blood (IFN-γ). These data were confirmed via histopathology with a positive effect on the inflammatory infiltration and myelin recovery [60].

Autoimmune encephalomyelitis is a brain inflammation originating from the body’s immune system, which attacks healthy cells and tissues in the brain or spinal cord and is sometimes misdiagnosed as a psychiatric or neurological disorder. It is a rare but complex disease, which can cause quick alterations of physical and mental health. Certain forms may be associated with tumors, such as ovarian tumors. In a mouse model of autoimmune encephalomyelitis, EXs derived from human MSCs and produced after stimulation with IFNγ were able to reduce neuroinflammation, decrease demyelination, and upregulate the amount of T cells characterized by CD4, CD25 surface antigens, and transcription factor forkhead box P3 within the spinal cords of treated mice [61]. Co-culture of the EXs with activated peripheral blood mononuclear cells in vitro reduced the cells proliferation and decreased pro-inflammatory cytokines levels, while immunosuppressive cytokine levels were increased. Anti-inflammatory RNAs were detected in the EXs [61].

#### 4.1.5. Spinal Cord Injury

Spinal cord injury is damaging to any part of the spinal cord or nerves that sends and receives signals from the brain to and from the rest of the body. It can be caused by direct injury or from damage to the tissue and vertebrae of the spinal cord, and it can be a temporary or permanent loss of sensory–motor function. A study evaluated the efficacy of miRNAs isolated from EXs in an experimental spinal cord injury animal model. Furthermore, studies on EXs produced in the presence of insulin growth factor-1 (IGF) were evaluated both in vitro and in vivo. High levels of miR-219a-2-3p were found in EXs stimulated with IGF, which led to the inhibition of the NF-κB pathway, a partial decrease in neuroinflammation and a promotion of neuroprotective effects in the animal model. The study evidenced that miR-219a-2-3p could be useful to decrease apoptosis and neuro-inflammation [62].

In a further study using EXs derived from MSCs, it was found that miRNA-216a-5p represents the most important miRNA to restore traumatic spinal cord injury by acting on microglial M1/M2 phenotypes. TLR4 was identified as the target downstream gene of miR-216a-5p together with NF-κB/PI3K/AKT signaling cascades [63].

EXs derived from human placental stem cells have proven to increase the expression of nerve trunk/progenitor cell markers in the spinal cord and to endorse recovery of both motor and autonomic functions after spinal cord injury experimental animal model with complete transection of the thoracic segment when intravenously administered. EXs also endorsed the stimulation of proliferating endogenous neural stem/progenitor cells and exhibited a significant high neurogenesis. In vitro, EXs promoted the production of neural stem cells and the increase in the phosphorylated levels of CREB, ERK, and MEK [64].

#### 4.1.6. Traumatic Brain Injury

Traumatic brain injury occurs when an unexpected, exterior, physical attack injures the brain, and it represents one of the most common causes of disability and death in adults, with a vast array of injuries that happen to the brain—which can be focal or diffuse, originating from mild to severe injuries. EXs obtained from astrocytes, enriched with miRNA-873a-5p, decreased neuroinflammation and reduced neurological deficits in post-traumatic brain injury models. Promotion of microglial polarization into the M2 phenotype was evidenced, and the mechanism was related to the inhibition of NF-*κ*B p65 and ERK phosphorylation [65]. In addition, EXs derived from astrocytes significantly decreased the mitochondrial levels of H_2_O_2_ and oxidative stress by the enhancement of the activity of antioxidant enzymes (principally SOD and CAT) in the neurons of the hippocampus of rats with traumatic brain injury [65]. In an in vitro study, EXs derived from bone marrow mesenchymal stem cells stimulated the polarization of activated BV2 microglia to induce the phenotype with anti-inflammatory activity. Studies in animals demonstrated the reduction in cell apoptosis in cortical tissue, the inhibition of neuroinflammation, and the promotion of synthesis of the microglia phenotype with anti-inflammatory activity [66]. Similar results have also been found for EXs derived from human adipose MSCs. These EXs enhanced functional recovery, decreased neuronal apoptosis, suppressed neuroinflammation, and enhanced neurogenesis in a rat model of traumatic brain injury. In addition, the inhibition of P38 mitogen-activated protein kinase signaling and NF-*κ*B was shown [67].

### 4.2. Rheumatoid Arthritis and Osteoarthritis

Rheumatoid arthritis (RA) is a chronic autoimmune disease characterized by inflammation of synovial tissue, leukocyte infiltration into the joints, liberation of inflammatory mediators and proteases, and degradation of the extracellular matrix mediated by matrix metalloproteinases (MMPs) [68]. A study demonstrated that EXs produced by MSCs, which overexpress miR-150-5p (Exos-150), can inhibit angiogenesis, migration, and invasion of fibroblast-like synoviocytes, which produce MMPs that degrade type II collagen and relieve the symptoms of RA by downregulating MMP14—which is considered a key enzyme mediating cartilage invasion and vascular endothelial growth factor (VEGF) expression [69]. Another strategy involves the modification of the surface of EXs via metabolic glycoengineering of the EXs produced by adipose-derived stem cells to target activated macrophages in the inflamed joints of RA. These EXs can reproduce natural inflammation resolution in the lesion via M1–M2 polarization, which defines a tissue remodeling and repairs improvement through the release of anti-inflammatory cytokine let-7b-5p, and miR-24-3p contained in EXs were responsible for M1–M2 macrophage polarization via the JAK-STAT signaling pathway [70]. Compared with the unloaded EXs, the loaded EXs reduced the level of IFN-γ and promoted macrophage polarization toward the M1 phenotype together with the secretion of TNF-α and IL-6 targeting the activated macrophages. Contemporarily, the IL-4 levels have been upregulated, promoting the macrophage polarization toward the M2 type, implying the depletion of the M1 macrophages [70]. Indeed, the imbalance between the activities of pro-inflammatory M1 and anti-inflammatory M2 macrophages in RA induces synovial inflammation and autoimmunity, leading to joint damage. During inflammation, M1-type macrophages produce pro-inflammatory factors, interleukin-1β (IL-1β) and IL-6, which promote RA progression. In contrast, M2-type macrophages secrete IL-10, a powerful anti-inflammatory agent, which inhibits the production of pro-inflammatory factors and blocks the inflammatory response while promoting healthy tissue remodeling. Therefore, increasing IL-10 secretion and decreasing pro-inflammatory factor levels can promote M1-to-M2 macrophage polarization, which can mitigate RA [71]. A study has evidenced that miR-204 synthetized by EXs plays an essential role in maintaining joint homeostasis to protect against RA pathogenesis. Indeed, exosomal miR-204-5p was present in mice with collagen-induced arthritis, while in humans, exosomal miR-204-5p expression was inversely associated with disease parameters of RA (rheumatoid factor, erythrocyte sedimentation rate, and C-reactive protein). The in vitro study demonstrated that human T lymphocytes produced EXs containing large amounts of miR-204-5p that were able to inhibit cell proliferation. In vivo tests demonstrated that the administration of lentiviruses expressing miR-204-5p noticeably relieved collagen-induced arthritis in mice [72].

Osteoarthritis (OA) is a common age-related degenerative disease of joints that causes disability in elderly individuals. It is characterized by synovial inflammation that affects all tissues of the joint and leads to a dysfunction of the whole joint. However, OA pathogenesis is not fully understood, and no regenerative treatment has been approved to prevent or slow down the disease progression [73]. The in vitro studies to evaluate the effects of EXs from different origins in regulating the repair and regeneration of chondrocytes are numerous. Among the principal mechanisms, EXs are able to promote chondrocyte proliferation likely by miRNAs, regulating gene expression and modulating various signaling pathways responsible for cell proliferation through the regulation of CDH11, NF-kB, ROCK1, TLR9, and Wnt5a [74,75,76,77]. Moreover, miRNAs from EXs produced by MSCs can inhibit chondrocyte apoptosis by preventing the expression of pro-apoptotic proteins (IL-1β, HMGB1, and RUNX2) [78,79,80]. Additionally, EXs produced by MSCs are able to stimulate the production of extracellular matrix components in chondrocytes, such as proteoglycans and collagen (principally aggrecan and collagen II). EXs produced by MSCs control the action of enzymes (aggrecanases and MMPs) involved in extracellular matrix synthesis and degradation, stimulating extracellular matrix synthesis and preventing degradation in chondrocytes [81,82].

An interesting study showed that EXs derived from MSCs can also be used as carriers to encapsulate curcumin, an anti-inflammatory drug with poor bioavailability. The in vitro modulatory effects of curcumin-primed human (h)BMSC-derived EXs (Cur-EXs) on IL-1β-stimulated human osteoarthritic chondrocytes were evaluated. Cur-EXs relieved IL-1β-induced catabolic effects on human osteoarthritic chondrocytes by endorsing viability and migration, decreasing apoptosis and phosphorylation of Erk1/2, PI3K/Akt, and p38 MAPK, thus modulating pro-inflammatory signaling pathways. Treatment of human osteoarthritic chondrocytes with Cur-EXs gives an upregulation of expression of hsa-miR-126-3p, which is involved in the modulation of anabolic responses of human osteoarthritic chondrocytes [73].

Another study tested the efficacy of EXs obtained from curcumin-treated, bone-marrow-derived MSCs in a surgically induced OA mouse model via intra-articular injection of the EXs, attenuating the progression of osteoarthritis and reducing the DNA methylation of miR-143 and miR-124 promoters. As a result, miR-143 and miR-124 were upregulated to further inhibit the expression of their target genes, ROCK1 and NF-kB, which were closely related to the development of OA [83].

A strategy to further treat osteoarthritis is the intra-articular injection of Platelet-Rich Plasma (PRP) in the joint because they are rich in EXs. Furthermore, to prolong the retention of EXs in the joints and obtain therapeutic effects, EXs were incorporated in injectable thermosensitive hydrogel (Poloxamer-407 and 188). In vivo studies demonstrated that EXs formulated in the gel improved their local retention, decreased the apoptosis and hypertrophy of chondrocytes, increased their production, and delayed the development of OA [84].

Intra-articular injection of EXs produced by MSCs derived from human embryonic cells improved gross appearance, upgraded histological scores, and gave the complete restoration of cartilage and subchondral bone [85]. In a similar study, the intra-articular injection of EXs produced by miR-140-5p overexpressing human synovial mesenchymal stem cells was tested in a surgically induced OA rat model. EXs improved the proliferation and migration of articular chondrocytes without harming extracellular matrix secretion and prevented OA in the rat model [86].

A rat model of osteoporosis was investigated to further characterize the beneficial effects of EXs derived from the MSC and, in particular, evaluate their safety profile. Indeed, the EXs from MSCs improved bone restoration and angiogenesis in a dose-dependent way [87].

The effect of EXs from bone marrow MSCs on OA was evidenced by an in vivo study using an OA rat model obtained via surgery. The intra-articular injection of both bone marrow MSCs or their EXs similarly protected the rat from joint damage. It was found that EXs preserved the chondrocyte matrix by enhancing collagen type II synthesis and reducing ADAMTS5, MMP13, and Col II expression in the presence of IL-1β in vitro. In addition, EXs protect chondrocytes from apoptosis and senescence, playing a particular role in regulating cell proliferation in repair and regeneration challenges [88].

### 4.3. Lung Injuries

Asthma is the principal disease among lung injuries affecting 300 million people worldwide. It is a heterogeneous chronic inflammatory airway disease, involving many cells of innate and adaptive immune systems. It acts on airway epithelial cells, triggering bronchial hyper-reactivity and airway remodeling in response to environmental stimuli such as allergens, infections, or air pollutants, which stimulate pro-inflammatory cytokine interleukins IL-4, IL-5, and IL-13 released by activated Th2 cells, increasing the number of airway mast cells and eosinophils. Additionally, mucus is observed [89]. A few studies have evidenced the association between circulating EXs and miRNAs mediated by EXs, being proposed as asthma biomarkers. Indeed, as a consequence of asthma, EXs are released from dendritic cells, eosinophils, mast cells, and bronchial epithelial cells and can modulate the chronic inflammatory processes involved in asthma [89]. A study evaluated the activity of EXs produced by M2 macrophages from semisynthetic origins, whose membrane was coated with PLGA@Dnmt3aossmart silencer (EM-PLGA@Dnmt3aossmart silencer) and administered intravenously to Der f1-induced asthma mice, a model of asthma allergies. EXs improved the pathology with a strong reduction in lung inflammation. EXs after the administration were distributed in various organs (including the lungs) and were able to target M2 macrophages. The treatment decreased the proportion of M2 macrophages and inflammatory cytokines in the airway, without suppressing the overall immune function [90]. Acute lung injury (ALI) and acute respiratory distress syndrome (ARDS) are further lung injuries characterized by acute respiratory failure with considerable morbidity and mortality. The incidence of these injuries is difficult to assess, and ALI/ARDS represents an unmet medical need. Both syndromes are characterized by the improved permeability of the membrane of alveoli and capillaries, interstitial and intra-alveolar edemas, severe hypoxemia, and a reduction in pulmonary compliance [91].

A recent review has evaluated the use of EXs to treat ALI/ARDS. The possible role of EXs can be related to their anti-inflammatory effects; the modulation of apoptosis and cell regeneration, particularly through the microRNA contained in exosomes, can participate in intercellular communication and play an immunomodulatory role in ALI/ARDS disease models [91]. The effectiveness of EXs produced by bone marrow MSCs in reducing the inflammatory factor expression and infiltration of neutrophils and macrophages was due to the miR-150. The EXs’ role was assessed by the regulation of the MAPK pathway [91]. In addition, EXs produced by MSCs decreased NF-kB activation and decreased the expressions of pro-inflammatory cytokines. Furthermore, EXs produced by MSCs decreased the expressions of proteins related to the TLR4/NF-kB pathway, and this effect could be reversed when the miR-451 expression in the EXs was inhibited. Additionally, EXs derived from MSCs improved the reduction in PI3K and AKT phosphorylation produced by oxidative damage in rat lung tissue due to the activation of the PI3K/AKT signaling pathway [91]. Additionally, EXs derived from bone marrow MSCs stimulated the expressions of anti-apoptotic proteins and ligand proteins by activating the Hippo-YAP pathway and JAK1-STAT3 signaling pathway, while EXs produced by airway myeloid-derived regulatory cells enhanced the T cell reaction in chronic inflammation by transferring mitochondria to T cells [91]. EXs produced by MSCs improved the production of reactive oxygen species and mitochondrial DNA damage and relieved mitochondrial dysfunction, restoring the barrier integrity in alveolar epithelial cells [91].

A very interesting study investigated the use of EXs produced by MSCs in preclinical models of ALI/ARDS. EXs produced by human umbilical cord MSCs reduced the levels of inflammatory factors and inhibited lung inflammation and oxidative stress in LPS-induced ALI by inducing autophagy in vivo. The study proved that miR-377-3p in the EXs played a crucial role in regulating autophagy. In particular, EXs released by human fetal lung fibroblast cells overexpressing miR-377-3p very efficiently inhibited the inflammatory factors and induced autophagy, resulting in the recuperation of ALI. The MiR-377-3p-targeted, regulatory associated protein of mTOR induces autophagy both in vivo and in vitro. [92].

EXs produced by endothelial progenitor cells were investigated in an ALI model of mice. EXs were able to reduce cytokines/chemokines and protein concentration in the bronchoalveolar lavage fluid, myeloperoxidase activity, pulmonary edema, and lung injury score due to the presence of miRNA-126 [93]. A further study confirmed that the EXs produced by the endothelial progenitor cell improved the endothelial cell function through miRNA-126 via the RAK/ERK signaling pathway and thus improving [94]. EXs obtained via alveolar progenitor type II cells promoted the production of ATII cells and improved epithelial regeneration in damaged alveolar cells in bleomycin-induced lung injury models. In particular, miRNA-371b-5p found in the EXs was responsible for the activity via Pi3k/Akt signaling [95].

A study evidenced that EXs produced by endothelial cells were able to inhibit topoisomerase II α expression, with a consequently protective role in a mouse model of sepsis-induced ALI. Activity was related to miRNA-125b-5p [96].

EXs produced by adipose tissue reduced the permeability of the pulmonary endothelial barrier and decreased the inflammatory reaction via the inhibition of the transient receptor potential vanilloid 4/Ca^2+^ signaling pathway. Therefore, there is a reduction in the pro-inflammatory cytokines induced by mechanical ventilation, resulting in a protective effect in a ventilator-induced lung injury [97].

A very interesting study investigated EXs engineered with a receptor for advanced-glycation-end-product-binding peptides, an anti-inflammatory peptide. The efficacy of these EXs was assessed using cytokine assays in lipolissacharide-activated macrophage cells. EXs were loaded with curcumin, which demonstrated that they had greater anti-inflammatory effects than curcumin alone or unloaded EXs. In the in vivo study, the EXs were administrated via intra-tracheal instillation into the lungs of the ALI model. A powerful reduction in pro-inflammatory cytokines and inhibition of inflammation were stated [98].

EXs have been also evaluated in patients with severe Coronavirus disease (COVID-19) who experienced acute respiratory distress syndrome (ARDS), as reported by a very recent review [99]. COVID-19 is an infectious disease caused by the SARS-CoV-2 virus, which can evolve to severe symptoms and, in some cases, death. A severe form of COVID-19 is characterized by lymphopenia and thromboembolic complications, disorders of the nervous system, kidney, and acute cardiac and liver injuries. Signs of inflammation and increasing levels of C-reactive protein and interleukin-6 are characteristic of severe patients [99]. WHO has calculated that about 800,000,000 patients experienced COVID-19, and about 7,000,000 patients died. Several studies have been reported to evaluate the activity of EXs produced from immune cells in reducing the inflammatory response and inhibiting the proliferation and activity of T helper cell 17, which has a significant role in inducing respiratory failure through the delivery of regulatory miRNAs [99]. One of the first studies involved three COVID-19 patients with severe ARDS and multi-organ complications, which were treated with Zofin (also known as Organicell Flow). The Zofin treatment resulted in a safe and well-tolerated therapy. Zofin was based on EVs that were naturally produced by human amniotic fluid. The patients showed improvements of the disease with a strong improvement of the inflammatory biomarkers [100].

In a further clinical trial, EXs produced by bone marrow MSCs were given intravenously for the treatment of patients with ARDS caused by severe COVID-19. The study demonstrated an improvement of oxygenation status and clinical symptoms of the improved respiratory function. In particular, the expressions of the inflammatory biomarkers (C-reactive protein and IL-6) reduced as well as decreased the Sequential Organ Failure Assessment scores. The decrease in ferritin and D-dimer was significant, and the neutrophil and lymphocyte cells grew without any adverse events [101].

In a very recent clinical study, ExoFlo (an extracellular signal product isolated from human bone marrow MSCs that contains growth factors and EVs including EXs) has been used to treat moderate to severe ARDS in patients with severe COVID-19. No adverse events were reported during the clinical trial, and the treatment was useful to decrease mortality compared with placebo. Ventilation-free days improved after ExoFlo treatment [102].

Other clinical studies on ALI/ARDS are in progress or completed, as evidenced by the website ClinicalTrial.gov. Therapeutic approaches are represented by EXs produced by human MSCs for the treatment of ARDS or novel Coronavirus pneumonia caused by COVID-19. Studies are related to the evaluation of the safety profile and efficacy of EXs containing CD24 (a glycosylated protein linked to the cell surface through means of a glycosyl-phosphatidylinositol anchor) in patients with a COVID-19 infection and extracellular vesicle infusion obtained from bone marrow MSCs for the treatment for COVID-19-associated ARDS.

### 4.4. Liver Injury

Liver injury and liver failure are diseases characterized by the necrosis and apoptosis of liver cells, preventing the liver from performing normal synthetic and metabolic functions. The main causes are viruses, drugs, lipid deposits, and autoimmune reactions; encephalopathy, cerebral oedema, sepsis, renal failure, gastrointestinal bleeding, and respiratory failure are the principal complications [68]. Chronic liver inflammation can also lead to fibrosis and cirrhosis. At present, liver transplantation remains the most effective option for patients with severe conditions, but several limitations still exist, such as lack of liver donors, high surgical costs, post-transplant complications, and immune rejections. Nucleoside binding and the NLRP3 inflammasome are the newly discovered pattern recognition receptors involved in the pathogenesis of many liver diseases. NLRP3 activation induces the death of inflammatory cells and pyroptosis and simultaneously leads to the maturation and secretion of IL-1β, suggesting an involvement in the progression of autoimmune hepatitis (AIH), a chronic inflammatory disease in the liver with the potential to promote the development of liver fibrosis. EXs produced by MSCs overexpressing miRNA-233 can inhibit NLRP3 activation and, consequently, liver inflammation and cell death. Moreover, the use of miRNA-17 carried by EXs produced by MSCs reduces NLRP3 inflammatory bodies [68]. A recent review on mesenchymal-stem-cell-derived exosomes as a new therapeutic strategy for liver diseases reported that EXs derived from human umbilical cord MSCs ameliorated liver fibrosis by inhibiting both epithelial mesenchymal transition of hepatocytes and collagen production. At the same time, EXs produced by adipose-tissue-derived MSCs expressing miR-122 have shown the ability to improve therapeutic efficacy in liver fibrosis treatment. Targeting hepatic stellate cells (HSCs), this miRNA helps the regulation of genes such as P4HA1 and IGF1R, which have been shown to be involved in HSC proliferation and collagen maturation [103].

Liver fibrosis is a chronic liver disease with the presence of progressive wound healing responses caused by liver injury. Currently, there are no approved therapies for liver fibrosis [104]. EXs derived from human adipose mesenchymal stem cells (hADMSCs-EXs) have been investigated for the anti-fibrotic effects both in vitro and in vivo. EXs inhibited the proliferation of activated hepatic stellate cells via apoptosis and by blocking mytosys in the G1 phase, efficiently preventing the expression of profibrogenic proteins and the epithelial-to-mesenchymal transition as demonstrated by in vitro studies. EXs were effective via the PI3K/AKT/mTOR signaling pathway [105].

EXs produced by Human Wharton’s jelly MSCs were evaluated in a model of liver fibrosis. In addition, EXs enriched of miRNA-124-3p were also evaluated. Both EXs efficiently reduced collagen accumulation and inhibited the inflammation. The therapeutic effect of the EXs enriched with miRNA-124-3p was significantly higher than the effects of EXs produced by human Wharton’s jelly MSCs. MiRNA-124 downregulated STAT3, which has a critical role in liver fibrosis. Both EXs were able to switch the phenotype of splenic monocytes Ly6Chi from inflammatory to restorative [106].

A liver injury further from surgery is the hepatic ischemia-reperfusion. EXs produced by adipose-derived stem cells have been evaluated for their hepatoprotective potential in a rat model of hepatic ischemia-reperfusion injury. After EXs’ treatment, liver histomorphology and hepatocyte ultrastructure changes were enhanced. The EXs’ treatment significantly downregulated IL-6, interleukin-1β (IL-1β), and TNF-α levels and upregulated IL-10 levels. EXs displayed their hepatoprotective properties by preventing endoplasmic reticulum stress and inflammation [107].

In a recent review, the role of EXs in some liver diseases has been reported [108]. It was found that the levels of miRNA-122 were enriched in EXs after drinking alcohol, as well as the release of inflammatory mitochondrial DNA from hepatocytes. HepG2 cells treated with ethanol and overexpressing CYP2E1 (involved in alcohol metabolism) released a high number of EXs that were able to change macrophages to the M1 type inflammatory phenotype. Nonalcoholic fatty liver disease is a liver injury with an extreme fat deposition in hepatocytes, generally connected with metabolic syndrome and represents the most widespread chronic liver disease worldwide. EXs produced by hepatocytes under lipotoxic settings stimulate NLRP3 inflammasome, resulting in IL-1β and caspase 1 activation. Free fatty acids stimulate hepatocytes to produce EXs containing—among others—miRNAs, which regulate hepatic lipid homeostasis [108].

A study investigated the association of metformin (first-line treatment for diabetes) and EXs produced by Wharton’s jelly MSCs in the HepG2 cells insulin resistance model. The study disclosed EXs enhance the activity of metformin without needing to change metformin by reducing inflammatory cytokines (TNF-α, IL-1, and IL-6) and apoptosis [109].

An interesting study investigated the effects of EXs produced by MSCs and hepatocytes given in association with imipenem in a mouse model of sepsis. Sepsis was induced in C57BL/6 mice, obtained by cecal ligation and puncture. Both EXs plus imipenem controlled local and systemic inflammation and enhanced the populations of T regulatory cells. However, the mixture of EXs plus imipenem had the best performance in decreasing inflammation, conserving all T lymphocyte populations, decreasing liver injury, and finally growing the survival rate [110].

Acute liver injury is generally associated with reduced prognosis but still there is a very limited number of effective drugs. In a study animal models (cecal ligation puncture and lipopolysaccharide plus D-galactosamine) to induce sepsis-induced acute liver injury. EXs produced by MSCs significantly reduced acute liver injury and consequent death. It was found that also miRNA-26a-5p was able to protect hepatocyte death and liver injury. The study revealed that the activity was due to Metastasis Associated Lung Adenocarcinoma Transcript 1 (MALAT1), an RNA Gene [111].

EXs produced by bone marrow MSCs were investigated alone or in association with praziquantel a potential therapy for schistosomal hepatic fibrosis in experimental animals. A significant reduction in hepatic fibrosis, of hepatic granulomas (number and diameter), a decrease in NF-κB expression and an upregulation of proliferating cell nuclear antigen expression were found in animals treated with EXs and those treated with EXs plus praziquantel [112].

### 4.5. Kidney Injuries

Among kidney injuries, Chronic Kidney Disease (CKD) is an increasing worldwide health problem, which leads to kidney failure, cardiovascular disease, and premature death [113]. This disease is tied closely with diabetes and hypertension. CKD treatments are limited to slowing disease progression through blood pressure and diabetes control or avoiding drugs that can worsen renal function. In addition to CKD, acute kidney injury (AKI), also known as acute renal failure, is a sudden episode of kidney failure or kidney damage [114] or kidney transplantation and dialysis. In further review, the use of EXs produced by MSCs is shown to treat CKD.

Another recent review has reported the latest advances in the use of EVs as a therapeutic approach for kidney injuries. In particular, EVs produced by bone-marrow-derived MSCs represent the extensive source for AKI recovery and a decrease in CKD progression [115]. The reported studies used EVs and non-isolated EXs in AKI, nephrectomia, and unilateral ureteral obstruction via intravenous administration, renal intra-capsular injection, and intra-arterial injection. The enhancement of tubular cell proliferation, anti-apoptosis and reduced fibrosis in the long term, support of morphologic and functional recovery, reduced tubular atrophy, improved kidney function, reduction in inflammation and cell death, and decreased cell apoptosis and inflammation represent the main effects of the EVs.

Studies with Adipose-Derived Mesenchymal Stromal Cells (ADMSCs) are mainly based on EVs used via intravenous or intra-arterial injection in an AKI, DOCA-salt hypertensive model, or unilateral renal stenosis. These studies evidenced a reduction in apoptosis, oxidative stress and inflammation, prevention of kidney fibrosis inhibition of apoptosis, immunomodulation, recovery of intracellular ATP, preservation of mitochondria, increased cell proliferation, angiogenesis, and immunomodulation. A study used the Perinatal-Derived Mesenchymal Stem Cells (PDMSCs) of EVs administered intravenously in Sepsis-AKI, reducing inflammation and apoptosis.

Another study involved EVs produced by Kidney Progenitor Cells (KPCs). The administration of EVs via intravenous injection in a model of AKI (unilateral ischemia-reperfusion injury plus unilateral nephrectomy) improved kidney function and reduced ischemic damage. Two studies tested EVs from Human Liver Stem Cells (HLSCs) via intravenous injection. The first trial was conducted in a model of diabetic nephropathy and showed prevention and reversal of the progression of glomerular and interstitial fibrosis. The second study demonstrated that intravenous injection of the EVs in a glycerol-induced AKI improved kidney function and cell proliferation and reduced tubular necrosis. Additionally, several studies have been reported using EVs produced by Induced Pluripotent Stem Cells (iPSCs) via intravenous and intrarenal injection in unilateral and bilateral IRI. Reduced inflammation, inhibited cell apoptosis, antioxidant effects, enhanced angiogenesis and cell proliferation, inhibition of endoplasmic reticulum stress and apoptosis, improved kidney function, cell proliferation, decreased tubular injury, cell death and fibrosis, mitigation of fibrosis, and reduction in necroptosis were found in the studies [115].

An interesting study investigated the efficacy of EXs originating from bone-marrow-derived MSCs in a model of diabetic kidney disease, the most common CDK form. Indeed, hyperglycemia induces oxidative stress and reactive oxygen species, activating the renin–angiotensin–aldosterone system and profibrotic cytokines. EXs repressed the extreme infiltration by regulating the expression of the adhesion molecule ICAM-1. EXs were able to inhibit proinflammatory cytokine expression (e.g., TNF-α) and fibrosis, downregulated TGF-β1 expression, and exerted an anti-apoptotic effect [116]. In further study using a rat model of streptozotocin-induced diabetes mellitus, EXs prominently enhanced renal function and exhibited restoration of renal tissues, with substantial growth of LC3 and Beclin-1 and substantial reduction in mTOR and fibrotic marker expression in renal tissue. All previous effects were partially abolished by the autophagy inhibitors chloroquine and 3-MA [117].

EXs produced by adipose-derived MSCs decreased DKD in mice (reduced levels of serum creatinine, urine protein, and blood urea nitrogen) by reversing autophagy downregulation and suppressing podocyte apoptosis [118].

EXs produced by human umbilical cord MSCs, administered in an irreversible model of unilateral ureteral obstruction, alleviated kidney fibrosis and restored renal function (decreased creatinine and blood urea nitrogen) through inhibition of apoptosis, malondialdehyde, ROS, and ROS-mediated P38MAPK/ERK signaling pathways. In vitro, similar anti-fibrotic effects were also observed in TGF-β1-treated NRK-52E cells [119].

The efficacy of EXs produced by adipose-derived MSCs was also evaluated in a renal artery stenosis model. EXs stabilized the systolic blood pressure, reduced profibrotic gene collogen and TGF-β expression, and downregulated hypoxia marker HIF-1a. Interestingly, both EXs, EVs, and the adipose-derived treatments of MSCs had a similar efficacy in decreasing the expression of the fibrotic markers TGF-β and collagen-1 [120].

More recently, a study investigated the activity of EXs produced by human-induced pluripotent-stem-cell-derived MSCs after intravenous infusion. EXs alleviated kidney fibrosis, decreased inflammatory responses, and increased renal function in mice subjected to unilateral ureteral obstruction. The effects of EXs were increasing via SIRT6 and reducing β-catenin and its products Fsp1, PAI-1, and Axin2 [121].

EXs produced by adipose-derived MSCs reduced levels of serum creatinine, blood glucose, 24 h urinary protein, urine albumin-to-creatinine ratio, and kidney weight/body weight and suppressed kidney fibrosis in a diabetic nephropathy model of rats. EXs also inhibited levels of IL6 and promoted cell apoptosis. The results evidenced that miRNA-125a was at least partially responsible for the efficacy of EXs [122].

EXs produced by human-urine-derived MSCs were investigated to prevent kidney complications from type I diabetes in rats after intravenous injection. EXs decreased urine volume and urinary albumin excretion, relieved albuminuria, and suppressed the caspase-3 overexpression [123]. In a similar study, the same EXs were tested in a model of diabetic nephropathy. The study indicated that miRNA-16-5p was evidenced to recover podocyte injury [124]. The protective function of EXs produced from adipose-derived MSCs was evaluated in a mice (C57/BL6) model of sepsis-induced acute kidney injury obtained via a puncture. EXs were able to activate SIRT1, decreasing the reverse microcirculation disorders, inflammation, and apoptosis and to reduce mortality [125].

In a very interesting study, a glial-cell-line-derived neurotrophic factor was transfected into human adipose mesenchymal stem cells via a lentiviral transfection system. EXs produced from the transfected cells were evaluated in vivo in a unilateral ureteral obstruction mouse model and in vitro in human umbilical vein endothelial cells. EXs significantly decreased renal fibrosis score and peritubular capillary in the in vivo studies. In vitro studies revealed EXs exerted cytoprotective effects on human umbilical vein endothelial cells by enhancing SIRT1 signaling and increased levels of phosphorylated endothelial nitric oxide synthase [126].

### 4.6. Intestinal Bowel Disease (IBD)

IBD is a chronic and debilitating disease characterized by idiopathic mucosal inflammation of the gastrointestinal tract. It includes several diseases, namely Crohn’s disease (CD), which may cause harmful inflammation in any part of the gastrointestinal tract and ulcerative colitis, which generally affects only the mucosa of the large intestine and it is characterized by bloody diarrhea. Recent studies have shown that genetic and environmental factors, immune response abnormalities, intestinal barrier dysfunctions, and intestinal flora imbalances are related to the development of IBD [127,128].

IBD disease also commonly causes the death of intestinal epithelial cells, not only destroying the integrity of the barrier but also causing inflammation, increasing the levels of inflammatory cytokines, such as TNF-α. The intestinal mucosal inflammation brings profound changes including depletion of local nutrients, imbalances in tissue oxygen supply, and in the demand and production of high quantities of reactive nitrogen and oxygen intermediates [129]. IBD causes are unclear, but the pathology is believed to be associated with genetic, environmental, gut microbiota, and immune response factors [129], and although significant progress has been made in its treatment, there are still some patients who face difficulties in being cured using traditional methods.

EXs derived from human umbilical cord MSCs carrying miR-378a-5p were investigated in a model of colitis. EXs induced suppression of interleukin (IL)-18, IL-1β secretion, and Caspase-1 cleavage abolishing cell pyroptosis and protecting against colitis [130]. A recent review has examined the effects mediated by EXs and applications in inflammatory diseases of the digestive system [131]. EXs produced by dendritic cells controlled the immune response and avoided the progress of autoimmune diseases [131]. EXs reduced expression of the pro-inflammatory cytokines interferon (IFN)-γ, TNF-α, IL-17A, and IL-22 and IL-12 and increased the anti-inflammatory cytokine TGF-β [131]. EXs produced by human umbilical cord MSCs have been investigated in a murine model of inflammatory bowel disease (dextran sulphate sodium treatment). A reduction in macrophages and the inhibition of their IL-7 expression were found [132]. EXs produced by *Staphylococcus enterotoxin A*-treated dendritic cells regulated the release of inflammatory cytokines in a mice model of colitis (dextran sulphate sodium treatment) [133].

## 5. Discussion

In the recent years, an incredible number of new studies have tremendously improved the knowledge of EXs’ role in inflammatory diseases, also elucidating specific targets and functions. EXs are nanosized vectors produced by almost all the human cells, which are released by cells as vectors of intercellular communication (over short and long-distance), both in health and disease conditions. In vitro studies and preclinical trials have evidenced EXs can play fundamental roles in the incidence and development of inflammatory disorders of the different body organs. Current knowledge indicates that the role of EXs in inflammation and in inflammatory diseases is extensive and complex.

EXs contain numerous constituents, i.e., cytoskeletal and cytosolic proteins, nucleic acids, lipids, and many other bioactive components (cytokines, signal transduction proteins, enzymes, antigen presentation and membrane transport/fusion molecules, and adhesion molecules), and their specific composition is related to their progenitor cells. In Table 1, a list of the animal studies related to EXs as nanotherapeutic agents in inflammation, focusing on several different inflammatory diseases, neurological diseases, rheumatoid arthritis and osteoarthritis, intestinal bowel disease, asthma, and liver and kidney injuries, is reported. Selection of the studies reported in Table 1 was based on the accurately described and validated extraction, purification, and characterization of EXs. Some specific analytical methods for evaluating characteristic surface marker molecules (CD9, CD81, CD63, etc.) of EXs were BCA protein assays and Western blotting.

The activity of EXs is due to additive/synergistic effects of the different molecules they contain, and the pieces of evidence that EXs can be ideal delivery vehicles for overcome physiological and pathological membranes, including blood–brain barrier, are numerous. EXs’ activity can be related to modulation of apoptosis and cell regeneration and anti-inflammatory effects. EXs can selectively release their cargo characterized by numerous molecules with different functional roles involved in multiple signal pathway transduction and regulating mitochondrial function, mainly represented by a variety of growth factors and non-coding RNAs, principally miRNAs. Additionally, surface manipulation of EXs, using molecules from genetic/biological engineering or chemical nature, is occasionally reported to optimize the interaction with targeted cells, as well as the loading of drugs or natural senolytic constituents, or simply an association of EXs and drugs given as conventional dosage forms to enhance the activity has been studied. Preliminary studies suggest that drugs (co-loaded or given in association in conventional dosage forms) and senolytic agents can increase the therapeutic activity, but still, their synergies/additivities with EXs are not to establish appropriate doses and possible risks.

Truly, the EXs’ interaction with the cells and tissues modulates numerous signaling pathways, as reported in the preclinical studies. In brief, in animal models of neuroinflammation, EXs provoke a decrease in TNF-α, an increase in IL-10, and stimulation of M2 polarization of microglia, which allow the reduction in the expression of inflammatory proteins (IL-1β, IL-18, CXCL2, and CXCL10). EXs regulate the Rheb-mTOR signaling pathway, with a significant decrease in CD3^+^T and CD4^+^T cells. In addition, in the animal models of AD, EXs reduced Aβ oligomer expression and decreased Aβ plaque deposition, via enhancing the PINK1/Parkin-pathway-mediated autophagy. Finally, TLR on microglia is activated.

Administration of EXs in animal models of lung injuries provoked the reduction in TNF-α, IL-1β, IL-6, and IL-8, as well as the number of inflammatory cells. Furthermore, an increase in IL-10 and superoxide dismutase activity was shown, while the levels of H_2_O_2_, malondialdehyde, and markers of oxidative stress diminished. Furthermore, there was a reduction in IL-4, IL-5, and IL-13 and a growth of TGF-β expression and T regulatory cells. A regulation of the MAPK pathway and NF-kB, a decrease in proteins expression related to TLR4/NF-kB pathway, and the activation of PI3K/AKT, Hippo-YAP, and JAK1-STAT3 signaling pathways are also described.

Similarly, the administration of EXs in animal models of kidney injuries increased the T regulatory cells, leading to an upregulation of CD206, the M2 macrophage marker. Reduced oxidative stress had increased expression levels of SOD1, AOX1, SIRT1, and SIRT2, as well as a reduction in the expressions of iNOS, TNF-α, IL-6, and IL-1β, while IL-10 increased.

Administration of EXs in animal models of liver injuries significantly reduced expression of inflammatory cytokines (TNF-α, IL-1β, IL-2, IL-6, IL-10, IL-17 MCP-1, and TGF-β) and M1 macrophage marker proteins (CD68 and CD11c). Reduced expressions of fibrosis-related proteins, inflammatory chemokines, and tissue inhibitors of metallopeptidase-1 were also observed.

Studies in animal models of rheumatoid arthritis and osteoarthritis evidenced that EXs regulate immunological reactivity, inhibit apoptosis, and stimulate proliferation. EXs inhibit CXCL9 and downregulated VEGF and MMP-14. EXs may regulate T cell differentiation and proliferation and the generation of Tregs. EXs increase IL-10 and decrease the immune response as well as inhibit IL-17. EXs inactivate the NF-κB and increase IL1RN, IL6, IL10 and IL11, and osteogenic factors (ALP, RUNX2, BMP2, and BMPR). Finally, EXs induce M1 macrophage polarization into M2.

In animal models of intestinal bowel diseases, the expression of pro-inflammatory cytokines (IL-1β and IL-6) was significantly reduced, while levels of IL-10 were increased. EXs also reduced JAK1 and STAT1 phosphorylation via the JAK-STAT signaling pathway. The suppression of IL-7, TNF-α, IL-6, and NF-κB is also reported.

## 6. Concluding Remarks and Future Perspectives

The reported studies have evidenced that EXs could represent a novel cell-free therapeutic strategy for modulating both inflammation and immunomodulation and may provide a novel treatment approach for inflammatory pathologies. EXs can act as nanocarriers for delivery of specific natural cargos, and their surface can be decorated with specific targets. Besides an autologous treatment, EXs can also be loaded with synthetic or natural drugs to obtain drug delivery systems useful as therapeutic agents. Furthermore, EXs are superior to the synthetic nano-drug delivery systems (vesicles, micelles, and nanoparticles) in terms of lower immunogenicity, higher structural stability, and lack of toxicity.

Although a significant number of preclinical studies have been reported in the literature, there is a scarcity of research concerning the possibility of translation to clinical applications. Clearly, treatments of EXs are superior to cell allogeneic transplantations because of poor cell survival rates and immunogenicity in vivo. However, there is an urgent need to know how the different EXs’ cargos are involved in the mechanism and how the scope of the diverse effects of EXs is produced from different sources. It is imperative to better understand their mechanisms of action in order to reduce the risk of side effects and increase their success rate. In the few clinical trials ongoing with EXs, a better pharmacokinetic profile, lower immunogenicity, and higher biocompatibility than synthetic nanomedicines have been evidenced. Results from animal models do not ensure the direct extension to humans.

Other EXs’ main limitations are related to their production, isolation, and storage. In particular, isolation methods must be refined in terms of yields and specificity. Presently, there is a lack of unique molecules to serve as biomarkers; indeed, EXs’ quality control is essential for safe and efficacious treatments. Clearly, EXs’ standardization should be easier than the whole EVs’ secretion.

Additionally, there is an urgent need to personalize the treatments in terms of the selection of type of EXs from different origins, their dosages, and routes of administration. Further validation in the clinical setting and standardization of methods will be required to bring this promising application into practice.

Additional requirements to bring this promising next-generation therapeutic option into practice are the comprehension of EXs’ biodistribution after administration, type and incidence of adverse events, and long-term effects.

Nanomedicines are generally approved according to the conventional framework, which is unfavorable to the development of EXs, and it is essential to improve the national and international regulations and establish widely acceptable standards.

## Figures and Tables

**Figure 1 pharmaceutics-15-02276-f001:**
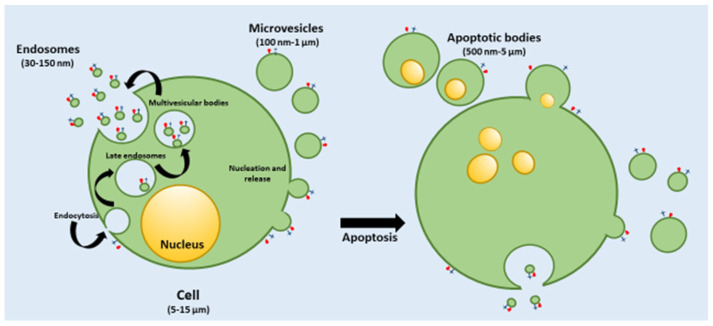
Production of extracellular vesicles: exosomes (EXs), microvesicles (MVs), and apoptotic bodies.

**Figure 2 pharmaceutics-15-02276-f002:**
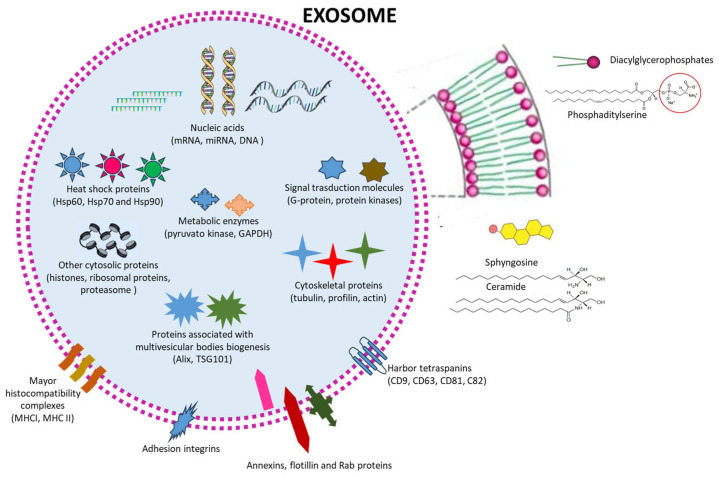
Structure of EXs, components decorating the membrane and constituents of the inner core.

**Table 1 pharmaceutics-15-02276-t001:** Studies in animals evidencing the role of EXs in in vivo inflammatory diseases.

Type of Disease/Animal Model	Exosome Origin (Eventual Drug Loaded)	Route of Administration	Effects	Reference
Neurological disorders/amyloid-β precursor protein transgenic mice	Neuroblastoma	Intracerebral injection	Marked reductions in Aβ levels and amyloid depositions.	[49]
Neurological disorders/transgenic APP/PS1 mice	MSC or hypoxia-preconditioned MSCs	Intravenously through lateral caudal vein	Both EXs, especially those deriving from the hypoxia preconditioned, improved learning and memory functions, reduced plaque deposition, and Aβ levels.	[50]
Neurological disorders/transgenic APP/PS1 mice	MSC	Intracerebroventricle injection	Reduction in exogenous Aβ-induced iNOS mRNA and protein expression; improved cognitive behavior.	[55]
Neurological disorders/PD model mice obtained with MPTP	MSC (EXs loaded with curcumin)	Intranasal administration	Reduction in α-synuclein aggregates, promoted neuron function recovery, significantly improved the movement and coordination ability.	[58]
Neurological disorders/middle cerebral artery occlusion mouse model	Hypoxic pre-treated, adipose-derived stem cells	Tail vein injection	Improvement of cognitive function by reducing neuronal damage in the hippocampus.	[59]
Neurological disorders/Experimental Autoimmune Encephalomyelitis	RAW (engineered EXs loaded with resveratrol)	Intranasal administration	Significant improvement of disease severity and significant decrease in multiple inflammatory cytokine responses in the CNS and periphery. Positive effect on the inflammatory infiltration and myelin recovery of MS.	[60]
Neurological disorders/Experimental Autoimmune Encephalomyelitis	MSCs (EXs stimulated by IFNγ)	Intravenous injections	Reduction in the mean clinical score and demyelination, decrease in neuroinflammation, and upregulation of CD4 + CD25 + FOXP3 + regulatory T cells.	[61]
Neurological disorders/spinal cord injury	Human placental MSCs	Intravenous injections	Significant improvement of the locomotor activity and bladder dysfunction; activation of proliferating endogenous neural stem/progenitor and a significative higher neurogenesis.	[62]
Neurological disorders/contusive spinal cord injury	MSCs under hypoxia	Tail vein injection	Promotion of functional behavioral recovery, shifting microglia from the M1 pro-inflammatory phenotype to the M2 anti-inflammatory phenotype by inhibiting TLR4/NF-κB and activating the PI3K/AKT signaling pathway.	[63]
Neurological disorders/spinal cord injury with complete transection of the thoracic segment	Human placental MSC	Intravenous injections	Enhanced anatomic and functional recovery; improvement of the locomotor activity and bladder dysfunction.	[64]
Neurological disorders/traumatic brain injury	Astrocyte	Tail vein injection	Mitigation of neurobehavioral deficits, cognitive impairment, and brain edema. Reduction in neuronal cell loss and atrophy, augmentation of antioxidant enzymes’ activities.	[65]
Neurological disorders/traumatic brain injury	Human adipose MSCs	Intracerebro-ventricular microinjection	Promotion of functional recovery, suppression of neuroinflammation, reduction in neuronal apoptosis, and increase in neurogenesis.	[67]
Rheumatoid arthritis and osteoarthritis/collagen-induced arthritis	Bone-marrow-derived MSCs	Intradermal injection	Reduction in hind paw thickness and decrease in clinical arthritis scores. Downregulation of MMP14 and VEGF expression, inhibition of angiogenesis.	[69]
Rheumatoid arthritis and osteoarthritis/collagen-induced arthritis	Adipose-derived stem cells (EXs with metabolic glycoengineered surface)	Intradermal injection	Induction of a cascade of anti-inflammatory events via macrophage phenotype regulation. High therapeutic efficacy, increase in T score, an indicator of bone density, and other clinical scores and photographs of inflamed joints.	[70]
Rheumatoid arthritis and osteoarthritis/collagen-induced arthritis	M2-type macrophages (EXs loaded with plasmid DNA encoding the anti-inflammatory cytokine interleukin-10)	Intradermal injection	Good accumulation at inflamed joint sites, high anti-inflammatory activity, and potent therapeutic effect.	[71]
Rheumatoid arthritis and osteoarthritis/osteoarthritis derived from injection of sodium iodoacetate	Bone marrow MSCs	Injection into the joint cavity	Effective promotion of cartilage repair and extracellular matrix synthesis, as well as alleviation of knee pain.	[82]
Rheumatoid arthritis and osteoarthritis/subtalar osteoarthritis by transecting anterior talofibular and calcaneal fibular ligament	Platelet-rich plasma (EXs were formulated in a thermosensitive hydrogel)	Injection in subtalar joint	Cartilage showed mild degeneration, and normal morphology and distribution of chondrocytes.	[84]
Rheumatoid arthritis and osteoarthritis/osteoarthritis model with critical-sized osteochondral defects from trochlear grooves of the distal femurs	MSCs derived from HuES9 human embryonic stem cells	Intra-articular injections	Complete restoration of cartilage and subchondral bone, including a hyaline cartilage with good surface regularity, complete bonding to adjacent cartilage, and extracellular matrix deposition.	[85]
Rheumatoid arthritis and osteoarthritis/osteoarthritis model from collagenase VII	Bone marrow MSCs	Injection	Inhibition of cartilage degradation and the progression of early osteoarthritis; prevention of the severe damage to knee articular cartilage.	[86]
Rheumatoid arthritis and osteoarthritis/osteoarthritis model with critical size bone defects in ovariectomized rats	MSCs from human induced pluripotent stem cells	Injection	Potent stimulation bone regeneration and angiogenesis.	[87]
Rheumatoid arthritis and osteoarthritis/osteoarthritis model by destabilization of the medial meniscus surgery	Bone marrow MSCs treated with decellularized extracellular matrix	Intra-articular injection	Better cartilage regeneration, improved anabolism and migration, and inhibiting chondrocyte apoptosis.	[88]
Lung injuries/Der f1-induced allergic asthma	M2 macrophages (EXs with coated PLGA@Dnmt3aossmart silencer)	Intravenous injection	Suppression of the development of asthma and marked reduction in inflammation, markedly decrease in the proportion of M2 macrophages and inflammatory cytokines.	[90]
Lung injuries/acute lung injury model with lipopolisaccharides	Human fetal lung fibroblast cells	Intratracheal instillation	Suppression of the bronchoalveolar lavage and inflammatory factors; induction of autophagy.	[92]
Lung injuries/acute lung injury model with lipopolisaccharides	Endothelial progenitor cell	Intratracheal instillation	Significant reduction in the cell number, protein concentration, and cytokines/chemokines in the bronchoalveolar lavage fluid. Reduction in myeloperoxidase activity, lung injury score, and pulmonary edema.	[93]
Lung injuries/acute lung injury model with lipopolisaccharides	Endothelial progenitor cell	Injection	Restoring pulmonary integrity, enhancing the proliferation, migration and tube formation of the endothelial cells.	[94]
Lung injuries/septic acute lung injury model via cecal ligation and perforation	Endothelial cells	Tracheal instillation	Promotion of VEGF expression, improvement of pathological changes, and control of lung water content, inflammatory response, protein content.	[96]
Lung injuries/ventilator-induced lung injury	Adipose-derived stem cell	Intravenous injection	Suppression of pulmonary endothelial barrier hyperpermeability, repair of the expression of adherens junctions, and alleviation of inflammatory response.	[97]
Lung injuries/acute lung injury model with lipopolisaccharides	Human Embryonic Kidney 293 cells (EXs engineered with RAGE-binding peptide, an anti-inflammatory peptide and loaded with curcumin)	Intratracheal instillation	Reduction in the cytokines in tissue lysate according to the levels of TNF-α and IL-1β in the bronchoalveolar lavage fluid. Reduction in hemolysis and infiltration of monocytes.	[98]
Liver injuries/carbon-tetrachloride-induced liver fibrosis	Human Wharton’s jelly MSCs (EXs enriched with miR-124-3p)	Intraperitoneal injection	Effective reduction in collagen accumulation and inhibition of inflammation. Phenotype switching of splenic monocytes from inflammatory Ly6Chi to restorative Ly6Clo.	[106]
Liver injuries/the portal vein and hepatic artery from 70% liver were occluded	Adipose-derived stem cells	Tail vein injection	Significant downregulation of TNF-α, interleukin-1β (IL-1β), and IL-6 levels; upregulation of IL-10 levels. In conclusion, ADSCs-exo protects against hepatic I/R injury after hepatectomy by inhibiting endoplasmic reticulum stress and inflammation.	[107]
Liver injuries/induced sepsis by cecal ligation and puncture	MSCs (EXs were co-administered with imipenem)	Imipenem injected subcutaneously plus EXs injected intravenously	Reduction in systemic and local inflammation; maintenance of all T lymphocyte populations, reduction in liver damage, and increase in the survival rate.	[110]
Liver injuries/induced sepsis by cecal ligation and puncture and lipopolysaccharide plus D-galactosamine	MSCs	Caudal vein injection	Replenishment of miR-26a-5p protected against hepatocyte death and liver injury caused by sepsis and inhibition of antioxidant system.	[111]
Liver injuries/schistosomal hepatic fibrosis	Bone marrow MSCs (EXs were co-administered with praziquantel)	Injection	Significant reduction in the number and diameter of hepatic granulomas, hepatic fibrosis; upregulation of proliferating cell nuclear antigen expression and reduction in NF-κB expression. Antifibrotic and anti-inflammatory effects.	[112]
Kidney injuries/diabetic nephropathy induced by streptozocin	MSCs	Injection	Marked improvement of renal function and restoration of renal tissues, with significant increase in LC3 and Beclin-1, and significant decrease in mTOR and fibrotic marker expression in renal tissue.	[117]
Kidney injuries/spontaneous diabetes	Adipose-derived stem cells	Tail intravenous injection	Attenuation of spontaneous diabetes by reduction in levels of urine protein, serum creatinine, blood urea nitrogen, and podocyte apoptosis.	[118]
Kidney injuries/renal fibrosis of irreversible model of unilateral ureteral obstruction	Human umbilical cord MSCs	Left renal artery injection after total ligation of the left ureter.	Significant decrease in the level of serum creatinine and blood urea nitrogen, of the level of apoptosis and oxidative stress. In addition, the renal tubular injury and tubulointerstitial.	[119]
Kidney injuries/renal chronic hypoxia induced by partial clamping of the left renal artery	Adipose-derived MSCs	Tail vein injection	Increase in stromal cell-derived factor-1-alpha (a cytokine) expression; reduction in the expression of hypoxia marker HIF1-α; stabilization of systolic blood pressure. Reduction in the expression of Col I and TGFβ and effective increase in the expression of the anti-inflammatory cytokine IL-10.	[120]
Kidney injuries/renal fibrosis from unilateral ureteral obstruction	Pluripotent stem-cell-derived MSCs	Tail vein injection	Reduction in the pathological process of renal fibrosis, of inflammatory reactions and improvement of renal function. Increasing the expression of SIRT6 and decreasing the expression of β-catenin and its downstream products.	[121]
Kidney injuries/diabetic nephropathy with streptozocin	Adipose-derived MSCs	Caudal vein injection	Decrease in levels of blood glucose, serum creatinine, 24 h urinary protein, urine albumin-to-creatinine ratio, and kidney weight/body weight. Suppression of mesangial hyperplasia and kidney fibrosis.	[122]
Kidney injuries/diabetic nephropathy with streptozocin	Urine-derived stem cells	Tail intravenous injection	Reduction in the urine volume and urinary microalbumin excretion, prevention of podocyte and tubular epithelial cell apoptosis, suppression of caspase-3 overexpression, and enhancement of the glomerular endothelial cell proliferation.	[123]
Kidney injuries/sepsis-induced acute kidney injury by cecal ligation and puncture	Adipose-derived MSCs	Tail vein injection	Activation of SIRT1, which reversed inflammation, apoptosis, and microcirculation disorders. In general, renal protective effects. Reduction in mortality.	[125]
Kidney injuries/renal fibrosis from unilateral ureteral obstruction	Glial-cell-line-derived neurotrophic factor transfected into human adipose MSCs via a lentiviral transfection system	Tail vein injection	Amelioration of renal fibrosis and significant decrease in peritubular capillary rarefaction Great capability of repair and angiogenesis.	[126]
Inflammatory bowel disease/colitis induced by dextran sulfate sodium	Human umbilical cord MSCs	Intravenous injection	Inhibition of NOD-like receptor family, pyrin domain-containing 3- inflammasomes of the colon. Suppression of the secretion of interleukin (IL)-18, IL-1β, and Caspase-1 cleavage, resulting in reduced cell pyroptosis.	[130]
Inflammatory bowel disease/colitis induced by dextran sulfate sodium	Human umbilical cord MSCs	Intravenous injection	Decrease in the severity of Inflammatory bowel disease. Increase in the expression of IL-10 gene and decrease in TNF-α, IL-1β, IL-6, iNOS, and IL-7 genes in colon tissues and spleens. Decreased of the infiltration of macrophages into the colon tissues.	[132]
Inflammatory bowel disease/acute colitis induced with dextran sulfate sodium	Dendritic cells treated with *S. japonicum* soluble eggs antigen	Intraperitoneal injection	Decrease in body weight loss and the disease activity index. Improvements of colon lengths and histological scores attenuating the severity of induced colitis in mice.	[133]

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
