# Peer review of "Exosomes: Potential Next-Generation Nanocarriers for the Therapy of Inflammatory Diseases"

_pharmaceutics, 2023, doi:10.3390/pharmaceutics15092276_

Round 1

Reviewer 1 Report

The manuscript by Mori et al summarized current status of application of extracellular vesicles for each diseases. They can further improve the manuscripts as described below.

Comments

1. The word EXs is confusing. Suddenly appears in line 71. Then in later paragraphs, EXs is often used but not sure if this EXs mean really Exosomes without MVs or apoptotic bodies. Usually each experiment cannot distinguish these things if it is only by size, so people use EVs if they are now sure if it is exosomes. The authors should clarify the usage of these words, EVx, EXs, MVs and apoptotic bodies. 

2. Exosomes derived from MSCs are often described in later paragraph. It is better to describe what is MSC in paragraph 2 or the beginning of 3. 

Author Response

  1. “The word EXs is confusing. Suddenly appears in line 71. Then in later paragraphs, EXs is often used but not sure if this EXs mean really Exosomes without MVs or apoptotic bodies. Usually each experiment cannot distinguish these things if it is only by size, so people use EVs if they are now sure if it is exosomes. The authors should clarify the usage of these words, EVx, EXs, MVs and apoptotic bodies. “

Thank you for this suggestion, all the reported studies were double-checked to be sure EXs and not EVs were used in the experiments. A table of the in vivo studies performed with EXs is reported (Table 1). In these experiments extraction, purification, and characterization of EXs is precisely described. All the steps are generally well-established and validated.  Transmission electron microscope represents only one of the analytical methods, which also include BCA protein assay kit and western blotting to evaluate the characteristic surface marker molecules CD9, CD81, CD63 etc... Indeed in many cases purified EXs were compared in terms of performance with EVs and/or MVs, these data are reported in the text. In addition a table concerning the solely data concerning the use and a table is added to clarify which type of EVs is used in each experiment

  1. “Exosomes derived from MSCs are often described in later paragraph. It is better to describe what is MSC in paragraph 2 or the beginning of 3.”

MSC and EXs derived from MSC are described in the new paragraph 2

Reviewer 2 Report

The present review evaluates the exosomes as potential next-generation nanocarriers for the therapy of inflammatory diseases. The topic is relevant, but certain deficiencies identified in both content and form need to be addressed based on the specific recommendations below:

1. The introduction section should be improved in terms of addressing the concept of nanomedicine and the advantages it brings over conventional medicine, as well as the possibilities of using nanosystems in both the diagnostic and therapeutic spheres (the concepts of passive, active targeted delivery or therapeutic action per se). I suggest checking and referring to: PMID: 37031724 and PMID: 33260701.

 2.The aim of the paper must be presented separately in the last paragraph of the introduction (section 1) and needs to be addressed from the perspective of describing the contribution to the field under analysis and the elements of scientific novelty presented.

 3. Section number two is very small, so it could be combined with the information in 1.1 for a very comprehensive section, especially as the introduction section does not need to be divided into several subsections.

 4.L165- 3 consecutive bibliographic indexes are represented a-c, a representing the first and c the last, please correct.

 5. L464 and L669- please correct the numbering

 6. Before the conclusions it is necessary to present in tabular form data from in vitro, in vivo (animal models) and clinical trials for each nanosystem used for the pathologies presented, so that their presentation is very clear, being much more difficult to follow only in text form.

 7. It is necessary to detail and highlight the applicability of nanomaterials and the implications they have for the management of pathologies. I suggest checking and referring to: https://doi.org/10.3390/chemosensors8040117.

 8. The conclusions section should be significantly reduced, leaving only essential elements, and the future perspectives section should be combined and improved in a new section with the limitations of nanotechnology, nanomedicine, nanocarriers.

Author Response

“The present review evaluates the exosomes as potential next-generation nanocarriers for the therapy of inflammatory diseases. The topic is relevant, but certain deficiencies identified in both content and form need to be addressed based on the specific recommendations below:

  1. The introduction section should be improved in terms of addressing the concept of nanomedicine and the advantages it brings over conventional medicine, as well as the possibilities of using nanosystems in both the diagnostic and therapeutic spheres (the concepts of passive, active targeted delivery or therapeutic action per se). I suggest checking and referring to: PMID: 37031724 and PMID: 33260701.”

Thank you for these suggestions, I introduced the passive and active targeting with a new literature by Couvreur and coworkers, he represents one of the father of the nanomedicine.

I also appreciated the suggestion of reading PMID: 37031724 (Radu AF, Bungau SG. Nanomedical approaches in the realm of rheumatoid arthritis. Ageing Res Rev. 2023 Jun;87:101927) and PMID: 33260701 (Chandra H, Singh C, Kumari P, Yadav S, Mishra AP, Laishevtcev A, Brisc C, Brisc MC, Munteanu MA, Bungau S. Promising Roles of Alternative Medicine and Plant-Based Nanotechnology as Remedies for Urinary Tract Infections. Molecules. 2020 Nov 28;25(23):5593. doi: 10.3390/molecules25235593). These two literatures from the same group of research are very interesting but I did not introduce them because not general but specifically related to rheumatoid arthritis or Urinary Tract Infections. The publication by Couvreur and coworkers was specifically related to the advanced nanomedicines for the treatment of inflammatory diseases.

 2.”The aim of the paper must be presented separately in the last paragraph of the introduction (section 1) and needs to be addressed from the perspective of describing the contribution to the field under analysis and the elements of scientific novelty presented.”

I did my best to prepare the new paragraph according to the precious suggestions of the reviewer

  1. “Section number two is very small, so it could be combined with the information in 1.1 for a very comprehensive section, especially as the introduction section does not need to be divided into several subsections.”

Text was modified accordingly

 4.”L165- 3 consecutive bibliographic indexes are represented a-c, a representing the first and c the last, please correct.”

Done

  1. “L464 and L669- please correct the numbering”

Done

 6.”Before the conclusions it is necessary to present in tabular form data from in vitro, in vivo (animal models) and clinical trials for each nanosystem used for the pathologies presented, so that their presentation is very clear, being much more difficult to follow only in text form.”

The table was prepared and added (Table 1). Only the in vivo studies in animal models were introduced because just a single study in humans is reported and in addition, the in vitro studies are numerous and their inclusion in the table can generate difficulties in interpreting the data. The in vitro studies were briefly discussed in the session of discussion

  1. “It is necessary to detail and highlight the applicability of nanomaterials and the implications they have for the management of pathologies. I suggest checking and referring to: https://doi.org/10.3390/chemosensors8040117.”

 Thank you for suggesting the paper “Nanomaterials for Diagnosis and Treatment of Brain Cancer: Recent Updates by Mahwash Mukhtar,Muhammad Bilal,Abbas Rahdar, Mahmood Barani, Rabia Arshad,Tapan Behl, Ciprian Brisc, Florin Banica  and Simona Bungau” specifically related to diagnosis and treatment of brain.

I added in the introduction of the manuscript some sentences concerning applicability of nanomaterials and the implications they have for the management of pathologies, I did not speak about diagnostic because aims of this manuscript are only related to therapeutics

  1. “The conclusions section should be significantly reduced, leaving only essential elements, and the future perspectives section should be combined and improved in a new section with the limitations of nanotechnology, nanomedicine, nanocarriers.”

A discussion section was introduced. Conclusions and future perspectives was rewritten. Limitations of nanocarriers (nanotechnology and nanomedicine) was added